



# The effect of salinity on the biogeochemistry of the coccolithophores with implications for coccolith-based isotopic proxies

Michaël Hermoso[1], Marceau Lecasble[1]

[1] Sorbonne Université, CNRS-INSU, Institut des Sciences de la Terre de Paris, 75005 Paris, France.

*Correspondence to*: Michaël Hermoso (michael.hermoso@sorbonne-universite.fr)

**Abstract.** Reconstruction of sea surface temperatures from the oxygen isotope composition ($\delta^{18}$O) of calcite biominerals synthesised in the mesopelagic zone of the oceans requires knowledge of the $\delta^{18}$O of seawater and constraints on the magnitude of biological $^{18}$O/$^{16}$O fractionation (the so-called vital effect). In the palaeoceanography community, seawater $\delta^{18}$O and salinity are unduly treated as a common parameter owing to their strong co-variation both geographically and in the
geological register. If the former parameter has arguably no notable influence on the biogeochemistry of marine calcifiers, salinity potentially does. However how salinity per se and the effect of osmotic adjustment can modulate the biogeochemistry, and in turn, the expression of the vital effect in calcite biomineral such as the coccoliths remains undocumented. In this culture-based study of coccolithophores (Haptophyta) belonging to the Noelaerhabdaceae family, we kept temperature and seawater $\delta^{18}$O constant, and measured basic physiological parameters (growth rate and cell size), and
the isotope composition ($^{18}$O/$^{16}$O and $^{13}$C/$^{12}$C) of coccoliths grown under a range of salinity comprised between 29‰ and 39‰. The overarching aim of this biogeochemical work has a geological finality and aims to refine the accuracy of palaeotemperature estimates using fossil coccoliths. We found that, although entailing large physiological changes in coccolithophores, salinity does not modulate biological fractionation in the oxygen isotope system. This is a contrasting observation with previous in vitro manipulations of temperature and carbonate chemistry that led to substantial changes in
expression of the vital effect. As such, salinity is not a complicating factor to derive temperatures from coccolith-bearing pelagic sequences deposited during periods of change in ice volume, especially at the highest latitudes, or in coastal regions. By contrast, the carbon isotope composition of the coccoliths is influenced by a growth rate-mediated control of salinity with implications for deriving productivity indices from pelagic carbonate.

## 1 Introduction

Salinity of the surface of the open oceans exhibits a wide range of values, typically comprised between 33‰ and 37‰, with more variability in coastal regions and in marginal seas (Antonov et al., 2010) (Fig. 1). Salinity has also significantly evolved in geological history, such as during the glacial/interglacial climate oscillations of the Plio-Pleistocene (Rostek et



al., 1993; Schmidt et al., 2004; Thornalley et al., 2009) and periods of emplacement of major ice-sheets in the Cenozoic (Zachos et al., 2001). Salinity is an important physico-chemical factor that is of great influence on pelagic communities, and as such, on the biological pump and the carbon cycle (Sarmiento and Gruber, 2006). This parameter co-varies with the oxygen isotope composition of seawater ($\delta^{18}O_{sw}$), both latitudinally and geologically (LeGrande and Schmidt, 2006) (Fig. 1).

This correlation is accounted for by a common control of the balance between evaporation and precipitation in a given area, and the volume of ice-caps at the highest latitudes. Therefore, salinity represents a signal of interest to constrain past environmental changes, but its variations is often approached as changes in $\delta^{18}O_{sw}$ values. Direct proxies for salinity reconstruction comprise e.g. the determination of hydrogen isotope ratios (D/H) of the alkenones, or Na/Ca elemental ratios in foraminiferal calcite (Schouten et al., 2006; Wit et al., 2013). Moreover, despite possible large changes in salinity

throughout an investigated time interval (e.g. glacial / interglacial cycles or longer-term studies), palaeoceanographers reconstructing sea surface temperatures (SSTs) from calcite $\delta^{18}O$ ratios do not fully integrate the possible modulation of the vital effect induced by salinity changes per se.

Perturbation of the living environment of the coccolithophores (single-celled photosynthetic Haptophyta microalgae with the

ability to biomineralise calcite intracellularly) has proven to significantly impact cellular growth rates and the geochemistry of the coccoliths in a wide range of isotopic and elementary systems (Bolton and Stoll, 2013; Candelier et al., 2013; Hermoso, 2015, 2016, Hermoso et al., 2015, 2016a, 2016b, 2017; Holtz et al., 2015; Katz et al., 2017; McClelland et al., 2017; Rickaby et al., 2016; Stevenson et al., 2014; Tremblin et al., 2016). Thus, our working hypothesis is that salinity changes will impact the physiology of coccolithophore cells, and as such, the expression of the vital effects, of which

modulation should be taken into account when applying isotopic proxies from coccolith-bearing sequences in the geological archive. To this aim, we cultured four strains of coccolithophore species in artificial medium with salinities from 29 to 39‰, monitored cellular growth rates and cell sizes, and measured the isotopic composition ($\delta^{18}O$ and $\delta^{13}C$) of the coccoliths.

## 2 Materiel and methods

### 2.1 Preparation of the culture media

We opted to conduct our cultures in artificial seawater (ESAW; Keller et al., 1987), as diluting natural seawater would have led to co-variations between salinity and $\delta^{18}O_{sw}$, and also because it is challenging to prepare medium with higher salinity than natural seawater due to the precipitation of salt occurring with evaporation. We thus maintained all ion ratios constant to better mimic natural environmental changes (Kirst, 1990). In total, we prepared two batches of 50 L each at salinity S = 39 (Sal39 thereinafter) by adding synthetic salts to de-ionised water, and subsequently prepared the lower salinity batches with

by re-diluting volumetrically the Sal39 batch with de-ionised water. We checked the salinity of each of the Sal29 to Sal39 batch by measuring the conductimetry of the media at room temperature. We used a TetraCon® 325 conductimetry cell



calibrated with 0.01 M KCl solution with a constant value of 1413 µS cm$^{-1}$ provided by the manufacturer (WTW Ltd). The correlation line linking conductimetry and salinity exhibited a root-square coefficient (r$^2$) of 0.99 indicating the good yield and accuracy of the targeted salinities. Therefore, in the following account, we will express the salinities of our batches by the target values, as they depart from less than 0.5‰ with respect to the actual measurements.

The various media were subsequently amended in nitrate, phosphate, silica and trace metals and vitamins as indicated in the *f*/2 recipe to ensure that macro and micronutrient availability did not change with salinity. Medium pH was adjusted to 8.2 (Total Scale) by addition of 1 N NaOH solution. The last step of the preparation consisted in steri-filtration of the artificial medium using 0.22 µm Millipore® Stericup® devices and the transfer of the medium into sterile culture vessels for further

algal inoculation.

**2.2 Strains selected for the bioassays**

We chose the following species and strains to conduct our experiments: *Emiliania huxleyi* morphotype A (strain RCC1256), *Emiliania huxleyi* morphotype B (strain RCC1212), *Gephyrocapsa oceanica* (strain RCC1314) and *Gephyrocapsa ericsonii* (strain RCC4032) (Fig. 1 for their sampling locations, and Fig. 2 for SEM images of their coccospheres). Originally, we

hoped to culture and test the effect of salinity on more species including *Calcidiscus leptoporus* (RCC1135), *Helicosphaera carteri* (RCC1323), and *Coccolithus pelagicus* (RCC1200), but the transfer into artificial medium (Sal35) was not successful, as the cells did not divide.

**2.3 Acclimatation phase**

The cultures were conducted at the biogeochemistry laboratory operating within the Institut des Sciences de la Terre de

Paris. In a first step, the cells were transferred from natural seawater originating from the English Channel to our Sal35 batch and left for a month (i.e. more than 20 generations). We sub-cultured the algae every week in fresh medium. Subsequently, the four strains acclimated in ESAW medium at Sal35 were transferred in each salinity, and the acclimatation phase to the new salinity was made for another month with the same sub-culturing frequency. The transfer of RCC1314 *G. oceanica* and RCC4032 *G. ericsonii* into the two extreme conditions Sal29 and Sal39, respectively, failed despite several attempts.

**2.4 Implementation of the batch cultures**

After the acclimatation phase, the proper bioassays begun with an initial concentration of cells around 200 cells mL$^{-1}$ of medium in 150 cm$^2$ polystyrene Thermo Scientific Nunc® culture flasks. Each strain/salinity condition was conducted in duplicate. The medium was refreshed every two days following a semi-continuous batch strategy. The cultures grew in a Panasonic® MLR 352 illuminated growth chamber. Temperature was kept constant at 15 °C, light irradiance was ~150 µmol

photons m$^{-2}$ s$^{-1}$ and the photoperiod lasted 14 hours per day. These flasks were gently stirred twice a day, and prior to



pipetting out 1 mL sterilely for cell numeration. Cell numeration was achieved using a Beckman® Coulter Counter® Multisizer IV fitted with a 70 μm aperture tube. The measurements were made three hours after the start of the illuminating period in the growth chamber. Calculation of the specific growth rates (μ) was made following Eq. (1):

$$\mu = \frac{\ln \ (\text{cell conc d}) - \ln \ (\text{cell conc d}-n)}{\text{number of days between d and d}-n} \tag{1}$$

where μ is the specific growth rate, *cell conc d* is the cell density in cells/mL at harvest and *cell conc d-n* is the cell density *n* day before this measurement. *Number of days between d and d-n* is the number of days elapsed between these two measurements (hence, *n* days). The unit of μ is day$^{-1}$.

The culture experiments were stopped when the densities reached 8,000 / 10,000 cells per mL. A final cell numeration was undertaken including the measurements of the size of the coccospheres (provided as a diameter-equivalent quantity by the Multisizer IV). The same aliquots were acidified by addition of 200 μL of 0.2 M HCl in the cuvette, left for 5 min and the measurements were repeated to provide the diameter of the naked cells. Removal of the coccospheres that originally surrounded the cells were checked under an inverted microscope Zeiss Axio Vert.A1® at 630 times magnification.

Standardisation of size measurements on the Multisizer IV was done via a calibration procedure that used certified latex beads *L10* with a 10.13 μm nominal diameter (batch 9747089F).

Harvest consisted in the suspensions containing the cells and free coccoliths to be poured onto polycarbonate membranes (0.8 μm nominal aperture) and gathered by vacuum-filtration. The culture residues were rinsed with deionised water to
prevent precipitation of salt during air-drying in a desiccating cabinet.

**2.5 Determination of isotopic ratios**

Aliquots of 0.22-μm filtered water samples were analysed for their $\delta^{18}O_{sw}$ using a PICARRO CRDS (cavity ring-down spectrometer; model L2130-I Isotopic H2O) at the LOCEAN-IPSL Lab in Paris. The $\delta^{13}C$ of the dissolved inorganic carbon (DIC) was measured from $CO_2$ (g) released from water after injection of phosphoric acid ($H_3PO_4$ 85%), and determined
using a dual inlet-isotopic ratio mass spectrometer (SIRA9-VG) in the same laboratory. These procedures and the standardisation are described in Racapé et al. (2010) and Benetti et al. (2017). The media were measured with $\delta^{18}O_{sw} = -6.55‰ \pm 0.1‰$ VSMOW, and $\delta^{13}C$ of the DIC with values of -12.40 ‰ ±0.3‰ VPDB. The isotopic values were indistinguishable between batches of distinct salinities (*n* = 12).

For coccolith calcite, approximately 100 μg of dry culture residue was weighted out and transferred into borosilicate vials. Calcite reacted with purified phosphoric acid at 70 °C in a Kiel IV preparation device attached to a Thermo Scientific Delta V Advantage isotope ratio mass spectrometer at ISTeP, Sorbonne University. Stable isotope values were calibrated relative





to the Vienna Pee Dee Belemnite (‰ VPDB) via the NBS-19 international standard. Reproducibility of measurements is ±0.1‰ for the $\delta^{18}O$ ratios and ±0.05‰ for the $\delta^{13}C$ ratios.

## 3 Results

### 3.1 Specific growth rates

All strains exhibit significant changes in their specific growth rates on the range of investigated salinities (Fig. 3). Overall, the relationship between salinity and µ can be described by second-order polynomial curves, with the exception of *G. ericsonii*, for which the measurements were statistically less well-behaved (partly due to the Sal31 measurements). *Gephyrocapsa oceanica*, the largest cell studied here grew faster at the lowest salinities Sal31 (0.52 day$^{-1}$), but surprisingly, it did not grow at Sal29. The µ for this strain progressively decrease down to 0.17 day$^{-1}$ with increasing salinity. The

relatively small cell of *G. ericsonii* shows the fastest division rate achieving more than one division per day for Sal33 and Sal35. The lowest salinities Sal29 and 31 exhibit µ values around 0.65 day$^{-1}$. They were significantly lower at the highest salinity Sal37, while reminding that no growth occurred at Sal39 for this strain.

The strain of *Emiliania huxleyi* producing coccoliths belonging to the Morphotype A (RCC1256) exhibits a bell curve for its

salinity/µ relationship ($r^2 = 0.91$) with a growth optimum at Sal33 (Fig. 3). On both sides, of the maximum value nearing 0.7 day$^{-1}$ (for reference, representing a doubling population of cells every day on average), the growth rates are significantly impaired with a symmetrical response. Indeed, Sal29 and Sal39 conditions exhibit halved growth rates relative to the optimum. The other strain of *Emiliania huxleyi*, RCC1212 Morphotype B, also exhibiting relatively smaller sizes, shows a different salinity/µ relationship, as growth rates did not significantly change and remained around 0.7 day$^{-1}$ with our different

salinity treatments, except for Sal29 that showed collapsed growth rates with values ~0.3 day$^{-1}$.

### 3.2 Cell sizes (diameters)

The strain RCC1314 of *Gephyrocapsa oceanica* exhibits an increase in cell diameter from ~6 µm to ~7 µm from Sal31 to Sal39 conditions (Fig. 4), which converted into volume, would correspond to an increase of the cellular volumes by 60% (116 µm$^3$ to 188 µm$^3$). This change can also be described by a second-order polynomial fit ($r^2 = 0.86$). The size of

*Gephyrocapsa ericsonii* is also highly dependent on salinity ($r^2 = 0.95$) with a progressive increase of the diameter and volume with increasing salinity, as it was observed for *G. oceanica*. We note, however, that the sizes for Sal35 and Sal37 are similar. For the small *G. ericsonii* cell, the relative variation in cell size is modest, less than 20% compared to its close-relative species *G. oceanica*.



No trend in size is evident for cells of RCC1212 *Emiliania huxleyi* grown at distinct salinities with diameter comprised between 5 μm and 5.6 μm (65 μm$^3$ and 95 μm$^3$ in volume). A relative variability exists for this strain, but is not related to ambient salinity (Fig. 4). The morphotype A of the same species has, in contrast, a sensibility of cellular volume to salinity, as these two parameters are statistically linked by a polynomial fit (r$^2$ = 0.81) with a minimum value at the intermediate

salinity (Sal 31-27 – 47 μm$^3$ on average), whereas Sal29 and Sal39 are characterised by higher cell volumes (87 μm$^3$ for Sal29).

The volume of coccolithophore cells is an important physiological parameter reflecting the metabolic rates of the cells (Aloisi, 2015). Such a relationship (the highest the growth rate, the smaller the cell size) is expressed for *G. oceanica*

(RCC1314) and the Morphotype A (RCC1256) of *E. huxleyi* (r$^2$ = 0.78 and 0.62, respectively) (Fig. 5), but is not for the two other strains, which do not exhibit responses describing a "bell curve" (second-order polynomial fit) (Fig. 4).

### 3.3 Isotopic ratios of coccolith calcite

### 3.3.1 Oxygen isotopes

A practical means to express the isotopic results from calcite minerals in biogeochemistry and palaeoceanography is to use the "δ – δ" notation, by which a pseudo-fractionation factor is expressed by the offset between δ$^{18}$O$_{calcite}$ and δ$^{18}$O$_{sw}$. Cultured coccoliths show a wide spread of δ$^{18}$O$_{calcite}$ – δ$^{18}$O$_{sw}$ values between species, from 0.5‰ for *G. oceanica* to 2‰ for *E. huxleyi* Morphotype A, but the variations for a given strain are much more limited, typically within a ~0.35‰ range (Fig. 6). Overall, there is no statistically-significant relationship between salinity and the oxygen isotope composition of the

coccoliths (r$^2$ < 0.6). It seems that coccoliths produced by *G. ericsonii* exhibit increase δ$^{18}$O values with increasing ambient salinity, but this correlation will not be regarded as strong, given the reproducibility of the replicates with respect to the range of measured values on the Sal29-37 spectrum.

In our study, the δ$^{18}$O$_{sw}$ did not significantly change with salinity, and temperature was kept constant at 15 °C, meaning that

changes in the isotopic composition of calcite correspond to a modulation of the expression of the vital effects by the coccolithophores. It is possible to compute the δ$^{18}$O composition of *an* inorganic calcite following work by Kim and O'Neil (1997). This inorganic reference has been recently found to depart from equilibrium conditions (Watkins et al., 2013), but all previous literature on coccolith biogeochemistry, including the assignment of coccoliths in an isotopic light and heavy groups (Dudley et al., 1986; Hermoso, 2014; Ziveri et al., 2003) are built on the Kim and O'Neil's reference. As our aim is to

compare the isotopic composition of coccolith calcite grown by microalgae exposed to various salinity, with once again, no change in temperature and δ$^{18}$O$_{sw}$, the use of this reference will not lead to an artefact for the biogeochemical signatures of the coccoliths. We calculated the oxygen isotope composition of such an inorganic calcite as –6.59‰ VPDB. The average



magnitudes of the oxygen isotope vital effect are thus +0.7‰ for RCC1314; +1.6‰ for RCC4032; +1.6‰ for RCC1212; +1.9‰ for RCC1256 – results that are in line with previously-published coefficients of the vital effect, except for RCC4032 *G. ericsonii* for which no documentation exists. For RCC1314 *Gephyrocapsa oceanica*, the magnitude of the vital effect is slightly lower, by 0.5‰, than previous reports with cultures of the same strain implemented in natural seawater (Hermoso et al., 2016a). This difference may be related to lower growth rate measured in our study at Sal 33-35 (0.45 day$^{-1}$ in artificial medium versus 0.85 day$^{-1}$ in natural seawater). In all cases, the important feature of this dataset is that we found no influence of salinity on the oxygen isotope vital effects.

### 3.3.2 Carbon isotopes

The spread of $\delta^{13}C_{calcite}$ – $\delta^{13}C_{DIC}$ offsets for a given strain is within 1‰ disregarding a non-replicated measurement for RCC1256 *E. huxleyi* at Sal 37 (Fig. 7). Both *Gephyrocapsa* cells (RCC1314 and 4032) show comparable trends for the $\delta^{13}C$ of their coccoliths with salinity, although the absolute values significantly differ (the former being 2‰ isotopically lighter than the latter). A negative trend is observed with increasing salinity, disregarding their lowest salinity datapoints. Such a trend is also evident for RCC1212 *E. huxleyi*, whereas for the other strain of *E. huxleyi*, $\delta^{13}C$ values are not well-behaved. Interestingly, the magnitude of these decreases in $\delta^{13}C$ values seems to be common to all species, and can thus be quantified to -1‰. For *G. ericsonii*, $\delta^{13}C$ and μ are strongly correlated by a polynomial function ($r^2 = 0.91$). This power fit is the mathematical consequence of the logarithmic expression of growth rate. Such a correlation is also seen, albeit statistically weaker, in RCC1212 *E. huxleyi* ($r^2 = 0.49$), but not for the two other strains.

For the carbon isotopes, the composition of the inorganic calcite is given by Romanek et al. (1992) and equals to $\delta^{13}C_{DIC} + 1$ after, hence -11.40‰ VPDB in our case study. Thus, *G. oceanica* appears to be close (±0.5‰) to inorganic conditions, whilst the other strains shift towards positive vital effects (up to +3.3‰ for RCC1212 *E. huxleyi*). These observations are also in good agreement with published literature (Hermoso, 2015; Hermoso et al., 2014; Holtz et al., 2015; McClelland et al., 2017; Rickaby et al., 2010). There seems to be, in contrary to the oxygen isotope system, an influence of salinity on coccolith $\delta^{13}C$ values, at least for three of the four strains being examined here.

## 4 Discussion

### 4.1 The effect of salinity change on the physiology of the coccolithophores

Seminal culture survey conducted by Brand (1984) unravelled the euryhaline nature of the coccolithophores *Emiliania huxleyi* and *Gephyrocapsa oceanica*, which have been successfully grown at salinities comprised between 25 and 45‰, with some strains of *E. huxleyi* exhibiting limited growth under salinity as low as 15‰. Our data and other previous studies

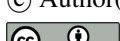


indicate that changing ambient salinity of the coccolithophores induce large changes of their growth rate (Fig. 8). In particular, it has been found that *E. huxleyi* and *G. oceanica* achieved better growth at the lowest salinities, with a lower limit of 25% for the former species (Fisher and Honjo, 1989; Saruwatari et al., 2016; Schouten et al., 2006). Except for Sal29, a salinity at which no growth was achieved by *G. oceanica* in our batches, our data for this species, and for *G. ericsonii*, are

compatible with this overarching observation that the cells exhibit diminished growth rate with increasing salinity (Fig. 3; Fig. 8). However, the salinity forcing on growth rate for the two strains of *E. huxleyi* being examined here markedly differs from observations made by Fisher and Honjo (1989) and Schouten et al. (2006), but are in line with those made by Fielding et al. (2009). An optimum in division rates at salinity around 33% (the mean oceanic value and the salinity at which the strains have been maintained in culture for many years) is evident in our data for *E. huxleyi* Morphotypes A and B. The

reason(s) behind this discrepancy are not straightforward to discuss beyond methodology, and may pertain to physiological differences and distinct adaptabilities to changing environment of the multiple ecotypes of *E. huxleyi* (e.g., Rickaby et al., 2016).

Our experiments only deal with short-term adaptability of the coccolithophores exposed to relatively abrupt changes in

salinity, and notably to rapidly adapt to the osmotic stress imposed to the cells in our treatments. The physiological implications regarding turgor stressor are complex, involving osmotic adjustment and the understanding of the precise mechanisms at play is beyond the scope of this paper. These aspects are not described in any great detail in existing literature. We see contrasting response in cellular sizes forced by ambient salinity among the strains grown in this study. *Gephyrocapsa oceanica* and *G. ericsonii* exhibit relatively smaller cell sizes at the lowest salinity (Fig. 5), while the opposite

response is observed for the Morphotype A of *E. huxleyi*. Meanwhile, there seems to be no response for the Morphotype B of *E. huxleyi*. From a physiological perspective, changing salinity changes the osmotic adjustment of the cells to maintain intracellular homeostasis, and thus, would influence the volume of the cytosol. Exposed to lowest salinities, unicellular eukaryotic cells are turgid as the consequence of a bigger water vacuole in the cytoplasm (Kirst, 1990). Such a salinity control on cell size is not compatible with our observations for *G. oceanica* and *G. ericsonii*, as at the lowest salinities the

cells are bigger, and *vice versa* (Fig. 4). For RCC1314 *G. oceanica*, the sizes of the cells are strongly correlated with growth rate. This is also the case for RCC1256 *E. huxleyi* (less clear for *G. ericsonii)*. These correlations between size and µ favour a control of cell size by the growth dynamics, which is itself modulated by salinity. As such, the "covariation of metabolic rates and cell size in coccolithophores" seems to be at play in our cultivated coccolithophores, a principle that stipulates that the more a cell population divides, the smallest the cells (Aloisi, 2015). The expression such an ecological process is not

obliterated by the large changes in salinity applied in the present culture study.

The preparation of our culture media did not only change the ionic concentrations of the dominant $Na^+$ and $Cl^-$, but also induced coeval compositional changes in elements relevant to photosynthetic and calcifying unicellular organisms. Indeed, from the Sal39 down to Sal 29 conditions, $[Ca^{2+}]$ and [DIC] decreased in the same ratio than salinity. Thus, a secondary




effect from simultaneous changes in $Ca^{2+}$ and DIC concentrations is therefore possible. For the lowest salinities, diluted $Ca^{2+}$ and DIC may become limiting for coccolithophore growth and intracellular calcification (calcium has also a role in cell signalling), superimposing a response by salinity per se. If the concentration and availability of DIC substantially decreased with decreasing salinity (constant 39:29 ratio), the availability of aqueous $CO_2$ in equilibrium with the lab air first and with

5 the headspace in the culture flask during the batch culture remained relatively unchanged. We calculated the variations in $[CO_{2aq}]$ using CO2CALC software (Robbins et al., 2010) and found indeed only relatively modest changes (on the order of 1%) from Sal39 to Sal29. Salinity has little effect on the $CO_{2air}$ / $CO_{2aq}$ equilibrium according to Henry's law, conversely to temperature. Furthermore, the linear changes in the concentrations of DIC, $CO_{2aq}$ and calcium on the entire salinity spectrum are not compatible with the bell curves observed in the measurements of coccolithophore physiology that exhibit optima

10 around Sal33-35. This question was also dealt with in work by Paasche et al. (1996) who conducted calcium and bicarbonate enrichment in low salinity waters, and found no alleviation of depressed growth rate. It is therefore unlikely that lowered growth rate at salinities lower than the optimum values is caused by decrease in the carbon and calcium resources. Therefore, we can confirm that the modulations of growth observed in our study are salinity-induced changes. Which ionic species $(Na^+, Cl^-, SO_4^{2-}...)$ exerts the foremost control of salinity on the coccolithophores at the cellular level remains an open

15 question.

## 4.2 The effect of salinity change on the isotopic composition of coccolith biominerals

### 4.2.1 Current knowledge of biological fractionation in coccolithophore calcite

Coccolithophores source DIC from the external environment to sustain growth (photosynthesis) and intracellular

20 mineralisation (calcification). The proportion of $CO_{2aq}$ and $HCO_3^-$ assimilated by the cells primarily depends on ambient $[CO_{2aq}]$, and the inducible expression of active mechanisms to uptake carbon in the form of $CO_{2aq}$ (excretion of carbonic anhydrase in the periplasmic environment) and $HCO_3^-$ (transmembrane transporters). Within the cells, the residence of the DIC pool between assimilation and biomineralisation of DIC in the coccolith vesicle, and the relative allocation of the carbon resource into photosynthesis relative to calcification have large implications on the isotopic composition of coccolith

25 calcite, both for the oxygen and carbon systems (Bolton and Stoll, 2013; Hermoso, 2014; Hermoso et al., 2016a, 2016b; Holtz et al., 2015; McClelland et al., 2017; Rickaby et al., 2010).

Specific to relatively small and fast growing species within the Noelaerhabdaceae family (a taxa to which our four strains belong), the coccoliths measured in our study have high $\delta^{18}O$ and $\delta^{13}C$ ratios compared to other species (see review in

30 Hermoso (2014)). These isotopic values are higher than inorganic conditions, and as such have been attributed to the isotopic "heavy group" (Dudley et al., 1986). Although transported by the same molecules and eventually measured from the same mineral, oxygen and carbon atoms have distinct behaviours in terms of dynamic of the two isotopic systems, and the




controlling factor dictating equilibrium condition to which is superimposed biological fractionation. These main differences include a large reservoir of atoms for the oxygen (DIC & ambient water molecules that constantly exchange $^{18}O$ and $^{16}O$); a sluggish re-equilibration time once a disequilibrium is created; and the temperature-dependence of fractionation between $^{18}O$ and $^{16}O$ between calcite and water (Zeebe and Wolf-Gladrow, 2001). For the carbon system, the size of the pool is markedly

smaller than for the oxygen. Photosynthesis favours $^{12}C$ at the expense of $^{13}C$ atoms, leaving the internal pool isotopically heavier, and in particular that that will be allocated to calcification. The extent to which calcite $\delta^{13}C$ will be pushed towards more positive values obeys to Rayleigh distillation driven by photosynthetic $^{12}C$ fixation, and the relative allocation of DIC into photosynthesis and calcification (referred to as the PIC:POC ratio [*particulate inorganic carbon & particulate organic carbon*]). With these biogeochemical processes in mind, isotopic heavy $\delta^{18}O$ values for *Gephyrocapsa* and *Emiliania*

coccoliths arise from fast growth and the biomineralisation of the predominantly-sourced $CO_{2aq}$ by the cells that occurs prior to full equilibration between $^{18}O$-excess-bearing-DIC and $H_2O$ (Hermoso et al., 2014, 2016a). Relatively high $\delta^{13}C$ of *Gephyrocapsa* and *Emiliania* calcite can be explained by the small size of these cells, and, overall, by low PIC:POC ratios, as POC exceed PIC in these fast growing species (more can be read in McClelland et al. (2017)).

### 4.2.2 Implications for the temperature proxy based on $\delta^{18}O$ composition

The $\delta^{18}O$ ratio of calcite remains the most widespread and quantitative temperature proxy in palaeoceanography. Assuming that constraints on the magnitude of the vital effect can be provided and also assuming constant $\delta^{18}O_{sw}$, a decrease by 1‰ correspond to an increase in temperature by 4 °C following thermodynamic predictions (Watkins et al., 2013). There are many factors, however, that can obscure the reliability of this $\delta^{18}O$–temperature relationship, such as the environmentally-driven change in the magnitude of biological fractionation.

Previous culture studies have highlighted large modulation of the expression of the vital effect under manipulation of the growth environment without changing the isotopic composition of the DIC or $\delta^{18}O_{sw}$ with implications for the temperature proxy. In particular, the concentrations of ambient $CO_{2aq}$ have proven to represent a major driver of the vital effect: The more availability of carbon around the cell, the less the vital effect (Hermoso et al., 2016b). Changing salinity induced

comparable changes in cellular growth rate comparable in magnitudes to those induced by manipulation of temperature and carbonate chemistry ([DIC] & pH) in previous culture studies of the coccolithophores. Yet, the vital effects are relatively stable for a given strain with temperature and $\delta^{18}O_{sw}$ being kept constant in our experiments (Fig. 6). When translated into temperature estimates, the most conservative error (averaging replicated measurements) from our $\delta^{18}O_{coccoliths}$ is 0.4 °C. As discussed in the previous section, coeval changes in the concentrations of $CO_{2aq}$ remain very limited. This important finding

pertaining the biogeochemistry of the coccolithophores indicates that changes in seawater salinity through geological history, as *e.g.* when studying glacial-interglacial changes at high latitudes or period marked by the emplacement of large continental ice-sheets, would have no impact on the reliability of the $\delta^{18}O$–temperature proxy analysing ancient coccoliths, at least for



those belonging the Noelaerhabdaceae family. More research would be needed in the future to test whether our conclusion could be extended to other geologically-relevant taxa.

### 4.2.3 Implications for the palaeoproductivity proxy based on $\delta^{13}C$ composition

The significance of $\delta^{13}C$ ratios of pelagic carbonates is not as clear as the $\delta^{18}O$ proxy in the palaeoceanographic toolbox. The
$\delta^{13}C$ tool depends on the timescale being considered with changes in $\delta^{13}C$ composition of seawater and carbonate being potentially due to rapid shifts in ocean circulation, upwelling, or drifts in primary productivity. As organic carbon has negative carbon isotope ratios relative to DIC, periods of increasing productivity and organic carbon export to the seafloor, are characterised by increased in carbonate $\delta^{13}C$, noting that this control resembles to processes operating at the intracellular level and dictating coccolith $\delta^{13}C$ values described above. An emerging research avenue utilises the differential expression of carbon limitation by large versus small coccolithophore cells as the new proxy for $pCO_2$ (Tremblin et al., 2016).

We see covariations between salinity, growth rates, and in turn, $\delta^{13}C$ values of the coccoliths cultured in our study, especially for the smaller cells. For *G. ericsonii*, $\delta^{13}C$ and $\mu$ are strongly correlated ($r^2 = 0.91$). The $\Delta\delta^{13}C$ variations of 1‰ on the range of investigated salinity are relatively important considering the natural variations of this isotopic tracer in geological past.
We would like to remind that the $\delta^{13}C$ of DIC did not change among batches of distinct salinity. The progressive decrease in coccolith $\delta^{13}C$ with increasing salinity can be the consequence of changing growth rates, but also could originate from the fact that less carbon is allocated to photosynthesis relative to calcification (meaning at first order that at the lower salinities, calcification is hampered). In the absence of measurements of PIC:POC ratios in our study (due to too low mass of residue harvested), it remains difficult to discuss these isotopic changes from a mechanistic point of view at the intracellular level.
We hope that the detailed salinity / $\delta^{13}C$ relationship will be scrutinised in the future, in particular to extend our understanding of the carbon isotope vital effects beyond $pCO_2$ and temperature.

### 5 Conclusion and future work

Open ocean microalgae, as coccolithophore microalgae, are confirmed to be euryhaline organisms able to thrive under salinities from 29 to 39‰, as they usually develop in waters of 33-35‰ salinity. Their tolerance to salinity changes and the induced modulation of their growth rates appear to be strain-specific, with an overall growth optimum around 33‰, the salinity of their natural environment. The physiological mechanisms and, in particular, those involved in the osmotic regulation of the coccolithophore cells grown under such a wide range of salinity are not well constrained, and more work is clearly needed from the biological side to achieve such an understanding.

From a geological proxy perspective, we showed that substantial, and rapid –*for timescales compatible with adaptation*– changes in ambient salinity has no impact on biological fractionation by the coccolithophores for the oxygen isotope system. As such, coccolith $\delta^{18}$O-derived sea surface temperatures are not compromised by this parameter that has yet proven to significantly affect algal growth. It is laudable that our study is extended to more species belonging to other more ancestral

families in forthcoming culture studies. Likewise, it would be extremely valuable to investigate the potential of Na/Ca ratios in coccolith calcite as proxy for ancient salinity (as done in foraminifera) and to conduct morphometric analyses of these minute biominerals with the same objective.

## 6 Data availability

All the data generated for this study are provided in Table 1.

## 7 Appendices

Table 1: Numerical dataset.

## 8 Author contribution

M.H. designed the experiments. M.L. and M.H. conducted the experiments. M.H. wrote the manuscript with inputs from M.L.

## 9 Competing interests

The authors declare no conflict of interest.

## 10 Acknowledgments

The data presented in the present study are from the Master's research project of M.L. under the supervision of M.H. We thank Nathalie Labourdette, Aïcha Namar and Gilles Reverdin for measurements of the isotopic ratios of carbonates and

waters. Ian Probert is warmly thanked for providing the strains grown in this study. We are grateful to Fabrice Minoletti for preparing the ODV maps shown in Figure 1 and for continuous support, and to François Baudin for making possible the use of the Multisizer IV apparatus. M.H. acknowledges financial support from the French *Agence Nationale de la Recherche* (ANR) – Project "CARCLIM" under reference ANR-17-CE01-0004-01 and from the Cluster of Excellence "MATISSE" led





by Sorbonne Université under reference ANR-11-IDEX-0004-02. Part of this work was also supported by the *Mission pour l'Interdisciplinarité* of the French *Centre National de la Recherche Scientifique* (CNRS) in the framework of the *Défi ISOTOP* with a grant awarded to Fabrice Minoletti and M.H. – Project "COCCOTOP". The setting-up of the culture facilities at Paris was made possible thanks to all these funding bodies and to the host Research Unit (ISTeP) via a *Coup de Pouce* Action.

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





**Figure 1: Map showing mean annual sea surface salinity (top) and the isotopic composition of seawater (bottom) and the sampling locations of cultured strains of coccolithophores in the present study. The map was generated using Ocean Data View package (Schlitzer, 2017) available at http://odv.awi.de.**





**Figure 2: Scanning electron micrographs of the four strains of coccolithophores (Haptophyta) cultured in the present study. Key to strain ID: RCC1314 is *Gephyrocapsa oceanica* (top left): RCC4032 is *Gephyrocapsa ericsonii* (top right); RCC1212 is *Emiliania huxleyi* Morphotype B (bottom left); and RCC1256 is *Emiliania huxleyi* Morphotype A (bottom right). Note the higher magnification for the smallest cells studied, namely RCC4032 (top right).**





**Figure 3: Effect of ambient salinity on the specific growth rates of coccolithophore algae. The data show a prominent modulation of growth rates, which appears to be strongly strain specific, with optimum growth achieved at salinities around 33%, except for**
*G. oceanica*. **All fits (red lines) correspond to a second-order polynomial law (not significant for RCC4032).**



**Figure 4: Effect of ambient salinity on the cell diameters of coccolithophore algae. As for growth rate, variations in the size of the cells (lightly decalcified coccospheres – see Methods) are species and strain dependent. All fits (red lines) correspond to a second-order polynomial law.**



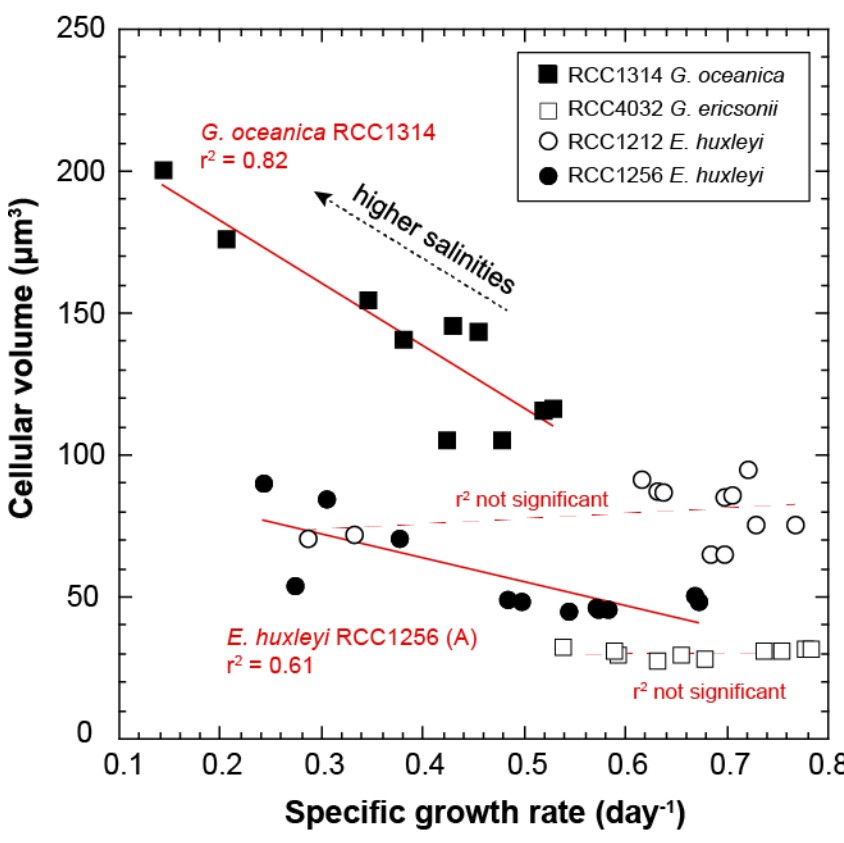

**Figure 5: Co-variations between growth rate and cell size of coccolithophores grown under various salinities. The conversion of the cell diameters into volumes have been achieved considering that the cells have are spherical. RCC1314 *G. oceanica* and RCC1256**

5    ***E. huxleyi* A show correlated changes in these two ecological-relevant parameters, however with distinct influence of salinity.**





**Figure 6: Effect of ambient salinity on the oxygen isotope composition of coccolith calcite. The variations in the isotopic composition of the coccolith are relatively minor, less than 0.35‰ for a considered strain with our salinity treatments. The spread of isotopic data is slightly larger for RCC1212 *E. huxleyi* (0.5 ‰), but there is no salinity dependence on growth rates observed for this strain. For reference, the Kim and O'Neil's calcite composition is around 0‰ VPDB. At 15°C, the vital effect can thus be quantified by the $\delta^{18}O_{calcite} - \delta^{18}O_{sw}$ offset. These positive values indicate that all the investigated species belong to the "isotopic heavy group". All fits (red lines) correspond to a second-order polynomial law.**





**Figure 7: Effect of ambient salinity on the carbon isotope composition of coccolith calcite. There is an influence of salinity, or a parameter itself influenced by salinity (growth rate) on the stable carbon isotope composition of the coccoliths. The is a general decrease in δ¹³C values with salinity, particularly well expressed on the Sal33-39 spectrum. RCC1256 _E. huxleyi_ shows appreciable spread in the data without a clear correlation between salinity and δ¹³C values.**




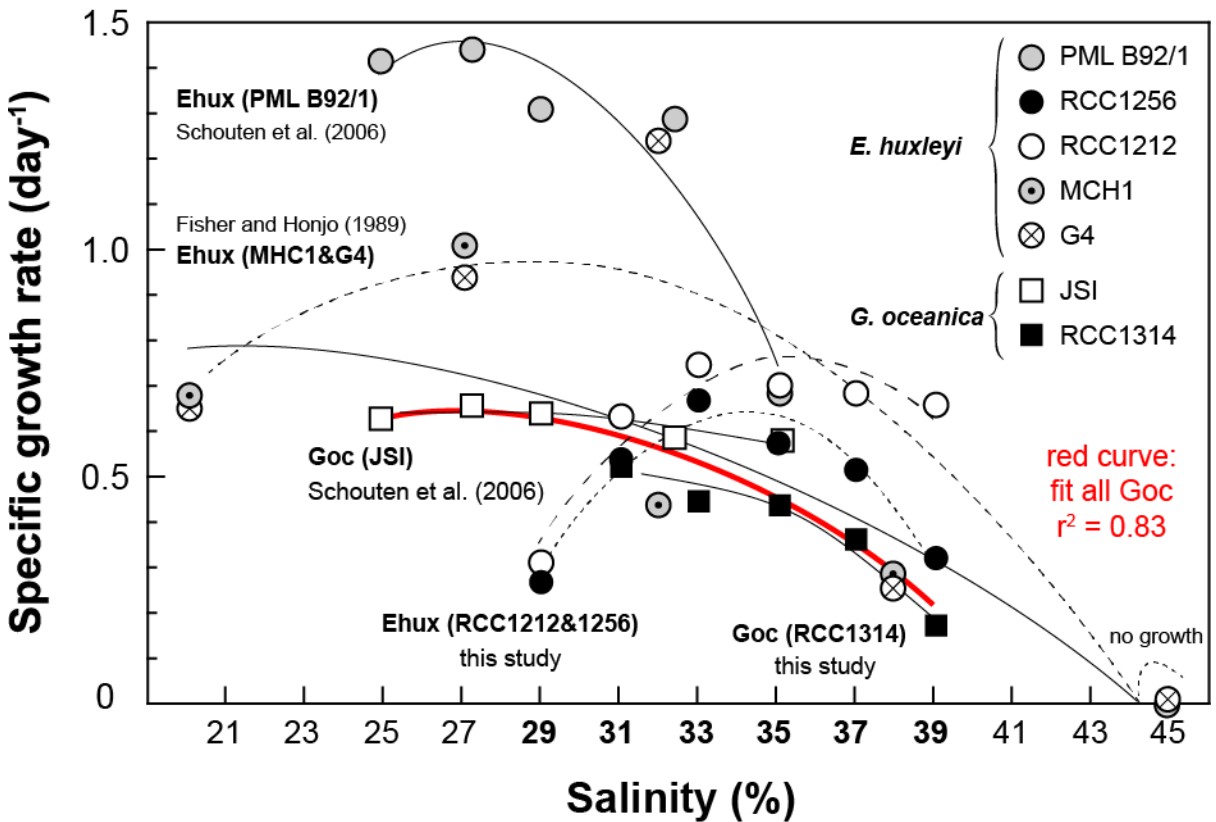

**Figure 8: Compilation of previously-published data documenting the variations of the specific growth rates of coccolithophores with salinity, and data of the present study shown on Fig. 3. The source for previous studies is inset. It is apparent that growth rates diminish with increasing salinity with a large variability in the response according to the considered strain, especially for *E.***
5 ***huxleyi*. By contrast, data for *G. oceanica* seem to be in good agreement between studies and strains (red curve), disregarding the fact that no growth was achieved by RCC1314 at Sal29 in our study. Overall, this graph highlights the euryhaline nature of open ocean microalgae that typifies the ecology of the coccolithophores thriving in the oceans and maintained in the laboratory under salinities nearing 33%.**

