# Peer review of "The effect of salinity on the biogeochemistry of the coccolithophores with implications for coccolith-based isotopic proxies"

_Biogeosciences, 2018_

## Referee Comment (RC1) · L.J. de Nooijer (Referee) · 16 Oct 2018

Review Hermoso and Lecasble (BG-2018-357)

Dear editor,

After careful assessment of the manuscript of Hermoso and Lecasble on the effect of salinity on coccolithophore calcite chemistry, I recommend publication of minor revisions. Below, I listed some minor comments that I hope will further improve the text. I have only one serious issue with this work: the statistical basis for the regressions is lacking. There is no explanation as why the authors chose second-order polynomal

fits or to what extent they fit the data better than linear functions. The authors often refer to '(in)significant' trends, but it remains unclear how this is determined. r^2 values are themselves no indication of significance, as the authors suggest. This needs to be addressed in the revised version of their manuscript.

SIncerely,

Lennart de Nooijer

Methods

page 2, line 26/27: evaporation of seawater by sub-boiling to a salinity of ~40 does not result in precipitation of salts. Preparing media by evaporation, however, also leads to differences in for example, total inorganic carbon concentrations, which might lead to differences in calcification. Perhaps the authors can use this as an alternative reason for preparing the culture media 'de novo'.

Which brings me to the question whether the carbonate chemistry (pH, [DIC], TA) of the water was determined/ monitored during the culturing experiment.

page 3, line 14-17: this can be omitted since it has no further relevance for this manuscript. Since this section will then become a bit short, it may better be combined with the previous or next one.

page 4, line 7: should be 'days'. I think in equation (1), the 'number of days between d and d-1' can be replace by 'n'.

Results

page 6, line 3: I don't understand the 'sensibility' in this sentence.

page 7, line 13: please avoid 'lighter', but instead use 'more depleted'. See also else-where.

page 7, line 15: please rephrase 'well-behaved'.

page 7, line 22: use 'equilibrium values' instead of 'inorganic conditions'.

Discussion

page 7, line 31/32 and later in the discsussion: salinity has no unit. Not even per mille.

page 9, line 4: but isn't the availability of $CO_2$ not also determined by the activity of the coccolithophores? E.g. by photosynthesis. Moreover, $CO_2$ may not be the preferred inorganic carbon species used for calcification (but maybe $HCO_3^-$), so I fail to see the logic of this argument.

page 11, line 10 and elsewhere: please italicize the 'p' in '$pCO_2$'.

Figures

I don't see the added value of figure 1. The spatial resolution is too coarse to see the sal/ del-18O of the water at the sampling locations. Otherwise, these maps show known global distributions in sal and del-18O and including them here therefore seems superfluous to me.

Figure 3, x-axis title and caption: salinity is unitless, so please remove the '%'. I think the upper x-axis title can also be removed. I find the $r^2$ values not very useful: they are by themselves not indicative of a significant correlation.

Figure 4: same as for figure 3. Please remove the trendline for E. hux, morphotype B: a trendline usually suggests a (significant) trend, which there is not in this case.

---

## Referee Comment (RC2) · Anonymous Referee #2 · 29 Oct 2018

Dear editor,

The work of Hermoso and Lecasble addresses the effect of seawater salinity on the oxygen isotope composition of coccoliths. This work was carried out on a variety of coccolithophores in laboratory controlled conditions (where temperature and d18O were constant). Physiological parameters and isotope composition were measured under a range of salinity. Hermoso and Lecasble have concluded that despite large physiological changes, salinity does not effect the oxygen isotope composition. This is an important observation as salinity may complicate the interpretation oxygen isotope composition in relation to sea surface temperatures.

[Figure]

Listed below are a few minor comments:

Line 29, Page 2 Can the authors clarify what they mean by 'synthetic salts'. Perhaps indicate the composition or recipe as this may change how an organism/cell responds to the salinity of its environment.

Line 11 and 26, Page 3 The use of the term 'bioassay' may be quite misleading. I would recommend using the term 'culture' or 'algal culture'

Line 28, Page 3 Include a reference for the method/protocol of semi-continuous batch culture/strategy. This wouldn't be apparent to someone who is unfamiliar with the methodology and is important if someone is thinking of repeating this experiment.

Line 7, Page 5 What do you mean by 'statistically less well behaved' (this is repeated again in Line 15, Page 7). This is quite subjective. I would suggest that the authors consider rephrasing this.

Line 7, Page 8 Consider including the strain names of the coccolithophores used by the different authors. This may explain the difference in the observed physiological response.

Line 14-30, Page 8 I don't think the authors can exclude the role of osmosis in determining the cell size/volume. Nor can they make the conclusion that cell size is determined by the metabolism of the organism alone. As the authors mention, there is little known about the process of osmoregulation in coccolithophores. As such it is worth considering the following:

1) The presence of active transporters (or membrane pumps) that may vary in type, numbers and work at different rates. This will naturally affect the transport/diffusion of water across the membrane, thus influencing the size/volume of the cell and leading to the observed difference between G. oceanica, G. ericsonii and E. huxleyi.

2) An organism may have different ways of maintaining water balance. For example, some organisms have the potential of varying their osmolytes (osmoadapta-
tion/osmoregulation). As such, the cellular content of G. oceanica, G. ericsonii and E. huxleyi may vary based on said ability.

Line 6, Page 10 Awkward turn of phrase. Perhaps remove 'that' (which has been repeated)

---

## Author Comment (AC1) · 29 Oct 2018

Dear Editor,

We are very grateful to the Reviewer 1 for his time in reading and reviewing our manuscript. We were very pleased to read his overall good opinion on our study. All his remarks can be relatively easily dealt with in a revised version of our manuscript, should we be invited to submit one. You will find our responses to his comments below.

Sincerely,

[Figure]

Michael Hermoso

————

*Review Hermoso and Lecasble (BG-2018-357)*

*Dear editor,*

*After careful assessment of the manuscript of Hermoso and Lecasble on the effect of salinity on coccolithophore calcite chemistry, I recommend publication of minor revisions. Below, I listed some minor comments that I hope will further improve the text. I have only one serious issue with this work: the statistical basis for the regressions is lacking. There is no explanation as why the authors chose second-order polynomal fits or to what extent they fit the data better than linear functions. The authors often refer to '(in)significant' trends, but it remains unclear how this is determined. r2 values are themselves no indication of significance, as the authors suggest. This needs to be addressed in the revised version of their manuscript.*

*SIncerely,*
*Lennart de Nooijer*

Authors's response: The choice of the second-order polynomial fits was not made *a priori*, but corresponds the best fit function of the dataset. The only implication of these "bell-curves" is that coccolithophores have salinity optima comprised in the range of the examined conditions, and that generally, growth rates (Fig. 3), cell size (Fig. 4), as well as the isotopic composition of certain coccolith species (Figs 6 and 7) changed on either sides of the optima. We will provide a Table as SI Material with the various regression coefficients for the polynomial (quadratic) and linear functions in each case.

The statistical issue regarding the only $r^2$ values presented in the text and figures will be addressed by performing suitable *goodness of fit* tests in a new version of the manuscript.

Methods

*page 2, line 26/27: evaporation of seawater by sub-boiling to a salinity of âĹij40 does not result in precipitation of salts. Preparing media by evaporation, however, also leads to differences in for example, total inorganic carbon concentrations, which might lead to differences in calcification. Perhaps the authors can use this as an alternative reason for preparing the culture media 'de novo'.*

Authors's response: This is a good point. We will add this and modify the rationale behind our choice to use the ESAW medium in the manuscript.

*Which brings me to the question whether the carbonate chemistry (pH, [DIC], TA) of the water was determined/ monitored during the culturing experiment.*

Authors's response: We opted to conduct our cultures using a very dilute (in terms of cell numbers) and semi-continuous batches to avoid artifacts linked to drifts in the composition of the medium. We did not monitor these parameters, as we refreshed the media every two days to ensure that the cells grew in the desired and initial conditions.

*page 3, line 14-17: this can be omitted since it has no further relevance for this manuscript. Since this section will then become a bit short, it may better be combined with the previous or next one.*

Authors's response: We will follow the Reviewer's suggestion.

*page 4, line 7: should be 'days'. I think in equation (1), the 'number of days be-tween d and d-1' can be replace by 'n'.*

Authors's response: We will replace this.

Results

*page 6, line 3: I don't understand the 'sensibility' in this sentence.*

Authors's response: We will change "sensibility" for "response".

*page 7, line 13: please avoid 'lighter', but instead use 'more depleted'. See also elsewhere.*

Authors's response: We agree. This will be changed.

*page 7, line 22: use 'equilibrium values' instead of 'inorganic conditions'.*

Authors's response: We will implement this change.

Discussion

*page 7, line 31/32 and later in the discsussion: salinity has no unit. Not even per mille.*

Authors's response: This is correct. We will change this throughout the body text and in the figures.

*page 9, line 4: but isn't the availability of CO2 not also determined by the activity of the coccolithophores? E.g. by photosynthesis. Moreover, CO2 may not be the preferred inorganic carbon species used for calcification (but maybe HCO3-), so I fail to see the logic of this argument.*

Authors's response: Most coccolithophore species acquire DIC predominantly in the form of aqueous $CO_2$ by passive diffusion through the cell membrane. It is true, however, that the concentration of DIC in the coccolith vesicle, *i.e.*, at the site of calcification, is made by active transport of $HCO_3^-$ from the cytoplasm - see papers by Kottmeier et al. (2014 - Photosynthesis Research) and McClelland et al. (2017 - Nature Communications) referenced in our manuscript. Our data are not able to resolve these biogeochemical features, as it will require knowledge of more information, as the relative allocation of the carbon resource into photosynthesis and calcification. These mechanistic considerations are interesting, but beyond the scope of this paper that we would like to keep at an empirical level. Nevertheless, we will clarify this point in our revised manuscript, as there is indeed one or two missing sentences required to disambiguate this.

*page 11, line 10 and elsewhere: please italicize the 'p' in 'pCO2'.*

Authors's response: We will do that.

Figures

*I don't see the added value of figure 1. The spatial resolution is too coarse to see the sal/ del-18O of the water at the sampling locations. Otherwise, these maps show known global distributions in sal and del-18O and including them here therefore seems superfluous to me.*

Authors's response: We can remove figure 1 indeed.

*Figure 3, x-axis title and caption: salinity is unitless, so please remove the 'percent'. I think the upper x-axis title can also be removed. I find the rËȨ2 values not very useful: they are by themselves not indicative of a significant correlation.*

Authors's response: We will follow all these recommendations and add the p-values.

*Figure 4: same as for figure 3. Please remove the trendline for E. hux, morphotype B: a trendline usually suggests a (significant) trend, which there is not in this case.*

Authors's response: We will implement these changes.

---

## Author Comment (AC2) · 30 Oct 2018

Dear Editor,

We are very grateful to the Reviewer 2 for the positive assessment of our paper. As detailed below in a point-by-point response to their comment, we will take into account all the remarks made during peer-review and modify the manuscript accordingly.

Sincerely,
Michael Hermoso

[Figure]

*Dear editor,*

*The work of Hermoso and Lecasble addresses the effect of seawater salinity on the oxygen isotope composition of coccoliths. This work was carried out on a variety of coccolithophores in laboratory controlled conditions (where temperature and d18O were constant). Physiological parameters and isotope composition were measured under a range of salinity. Hermoso and Lecasble have concluded that despite large physiological changes, salinity does not effect the oxygen isotope composition. This is an important observation as salinity may complicate the interpretation oxygen isotope composition in relation to sea surface temperatures.*

*Listed below are a few minor comments:*

*Line 29, Page 2 Can the authors clarify what they mean by 'synthetic salts'. Perhaps indicate the composition or recipe as this may change how an organism/cell responds to the salinity of its environment.*

Authors's response: We now explicitly refer to the ESAW "recipe" with reference to Keller et al. (1997), so that the chemical composition of the medium can be found.

*Line 11 and 26, Page 3 The use of the term 'bioassay' may be quite misleading. I would recommend using the term 'culture' or 'algal culture'*

Authors's response: We changed "biossays" for "cultures".

*Line 28, Page 3 Include a reference for the method/protocol of semi-continuous*

*batch culture/strategy. This wouldn't be apparent to someone who is unfamiliar with the methodology and is important if someone is thinking of repeating this experiment.*

Authors's response: Reference to the chapter written by LaRoche et al. (2010) in the seminal book "Guide to best practices for ocean acidification research and data reporting" has been added.

*Line 7, Page 5 What do you mean by 'statistically less well behaved' (this is repeated again in Line 15, Page 7). This is quite subjective. I would suggest that the authors consider rephrasing this.*

Authors's response: Sentence modified for: "...with the exception of G. ericsonii, for which the measurements do not show a relation between $\mu$ and salinity."

*Line 7, Page 8 Consider including the strain names of the coccolithophores used by the different authors. This may explain the difference in the observed physiological response.*

Authors's response: This has been done.

*Line 14-30, Page 8 I don't think the authors can exclude the role of osmosis in determining the cell size/volume. Nor can they make the conclusion that cell size is determined by the metabolism of the organism alone. As the authors mention, there is little known about the process of osmoregulation in coccolithophores. As such it is worth considering the following:*

*1) The presence of active transporters (or membrane pumps) that may vary in type, numbers and work at different rates. This will naturally affect the transport/diffusion of water across the membrane, thus influencing the size/volume of the*

*cell and leading to the observed difference between G. oceanica, G. ericsonii and E. huxleyi.*

*2) An organism may have different ways of maintaining water balance. For example, some organisms have the potential of varying their osmolytes (osmoadaptation/osmoregulation). As such, the cellular content of G. oceanica, G. ericsonii and E. huxleyi may vary based on said ability.*

Authors's response: We agree with the Reviewer and will simplify our text regarding these biological concepts and follow the Reviewer's first point.

Lines 21-30 on p8 have been deleted and replaced by the following statement, which mostly relies on the Reviewer writing: "The presence of active transporters (or membrane pumps) may vary in type, numbers and work at different rates in various coccolithophore species. As a consequence, these possible distinct strain-specific features will affect the transport/diffusion of water across the membrane, thus influencing the size of the cell and leading to the observed differences between G. oceanica, G. ericsonii and the two strains of E. huxleyi."

*Line 6, Page 10 Awkward turn of phrase. Perhaps remove 'that' (which has been repeated)*

Authors's response: This has been modified. Sentence now reads: "Photosynthesis favours $^{12}$C at the expense of $^{13}$C atoms, leaving the internal pool isotopically more positive with implications for the carbon pool that will be allocated to calcification."

---

## Author Response (AR1)

Responses to RW1

Dear Editor,\\
\\
We are very grateful to the Reviewer 1 for his time in reading and reviewing our manuscript. We were very pleased to read his overall good opinion on our study. All his remarks can be relatively easily dealt with in a revised version of our manuscript, should we be invited to submit one. You will find our responses to his comments below.\\
\\
Sincerely,\\
\\
Michael Hermoso\\
\\
---------\\
\\
\textit{Review Hermoso and Lecasble (BG-2018-357)}\\
\\
\textit{Dear editor,}\\
\\
\textit{After careful assessment of the manuscript of Hermoso and Lecasble on the effect of salinity on coccolithophore calcite chemistry, I recommend publication of minor revisions. Below, I listed some minor comments that I hope will further improve the text. I have only one serious issue with this work: the statistical basis for the regressions is lacking. There is no explanation as why the authors chose second-order polynomal  fits or to what extent they fit the data better than linear functions. The authors often refer to '(in)significant' trends, but it remains unclear how this is determined. r^2 values are themselves no indication of significance, as the authors suggest. This needs to be addressed in the revised version of their manuscript.} \\
\\
\textit{SIncerely,}
\\
\textit{Lennart de Nooijer}\\
\\
\underline{Authors's response:} The choice of the second-order polynomial fits was not made \textit{a priori}, but corresponds the best fit function of the dataset. The only implication of these "bell-curves" is that coccolithophores have salinity optima comprised in the range of the examined conditions, and that generally, growth rates (Fig. 3), cell size (Fig. 4), as well as the isotopic composition of certain coccolith species (Figs 6 and 7) changed on either sides of the optima. We will provide a Table as SI Material with the various regression coefficients for the polynomial (quadratic) and linear functions in each case. The statistical issue regarding the only r$^2$ values presented in the text and figures will be addressed by performing suitable \textit{goodness of fit} tests in a new version of the manuscript.\\
\\
\bold{Methods}\\
\\
\textit{page 2, line 26/27: evaporation of seawater by sub-boiling to a salinity of ~40 does not result in precipitation of salts. Preparing media by evaporation, however, also leads to differences in for example, total inorganic carbon concentrations, which might lead to differences in calcification. Perhaps the authors can use this as an alternative reason for preparing the culture media 'de novo'.}\\
\\
\underline{Authors's response:} This is a good point. We will add this and modify the rationale behind our choice to use the ESAW medium in the manuscript.\\
\\
\textit{Which brings me to the question whether the carbonate chemistry (pH, [DIC], TA) of the water was determined/ monitored during the culturing experiment.}\\
\\
\underline{Authors's response:} We opted to conduct our cultures using a very dilute (in terms of cell numbers) and semi-continuous batches to avoid artifacts linked to drifts in the composition of the medium. We did not monitor these parameters, as we refreshed the media every two days to ensure that the cells grew in the desired and initial conditions.\\
\\
\textit{page 3, line 14-17: this can be omitted since it has no further relevance for this manuscript. Since this section will then become a bit short, it may better be combined with the previous or next one.}\\
\\

Responses to RW1

\underline{Authors's response:} We will follow the Reviewer's suggestion.\\
\\
\textit{page 4, line 7: should be 'days'. I think in equation (1), the 'number of days between d and d-1' can be replace by 'n'.}\\
\\
\underline{Authors's response:} We will replace this.\\
\\
\bold{**Results**}\\
\\
\textit{page 6, line 3: I don't understand the 'sensibility' in this sentence.}\\
\\
\underline{Authors's response:} We will change "sensibility" for "response".\\
\\
\textit{page 7, line 13: please avoid 'lighter', but instead use 'more depleted'. See also elsewhere.}\\
\\
\underline{Authors's response:} We agree. This will be changed.\\
\\
\textit{page 7, line 22: use 'equilibrium values' instead of 'inorganic conditions'.}\\
\\
\underline{Authors's response:} We will implement this change.\\
\\
\bold{Discussion}\\
\\
\textit{page 7, line 31/32 and later in the discsussion: salinity has no unit. Not even per mille.}\\
\\
\underline{Authors's response:} This is correct. We will change this throughout the body text and in the figures.\\
\\
\textit{page 9, line 4: but isn't the availability of CO2 not also determined by the activity of the coccolithophores? E.g. by photosynthesis. Moreover, CO2 may not be the preferred inorganic carbon species used for calcification (but maybe HCO3-), so I fail to see the logic of this argument.}\\
\\
\underline{Authors's response:} Most coccolithophore species acquire DIC predominantly in the form of aqueous $CO_2$ by passive diffusion through the cell membrane. It is true, however, that the concentration of DIC in the coccolith vesicle, \textit{i.e.}, at the site of calcification, is made by active transport of $HCO_3^-$ from the cytoplasm - see papers by Kottmeier et al. (2014 - Photosynthesis Research) and McClelland et al. (2017 - Nature Communications) referenced in our manuscript. Our data are not able to resolve these biogeochemical features, as it will require knowledge of more information, as the relative allocation of the carbon resource into photosynthesis and calcification. These mechanistic considerations are interesting, but beyond the scope of this paper that we would like to keep at an empirical level. Nevertheless, we will clarify this point in our revised manuscript, as there is indeed one or two missing sentences required to disambiguate this.\\
\\
\textit{page 11, line 10 and elsewhere: please italicize the 'p' in 'pCO2'.}\\
\\
\underline{Authors's response:} We will do that.\\
\\
\bold{Figures}\\
\\
\textit{I don't see the added value of figure 1. The spatial resolution is too coarse to see the sal/ del-18O of the water at the sampling locations. Otherwise, these maps show known global distributions in sal and del-18O and including them here therefore seems superfluous to me.}\\
\\
\underline{Authors's response:} We can remove figure 1 indeed.\\
\\
\textit{Figure 3, x-axis title and caption: salinity is unitless, so please remove the 'percent'. I think the upper x-axis title can also be removed. I find the $r^2$ values not very useful: they are by themselves not indicative of a significant correlation.}\\
\\
\underline{Authors's response:} We will follow all these recommendations and add the p-values.\\
\\

**Responses to RW1**

\textit{Figure 4: same as for figure 3. Please remove the trendline for E. hux, morphotype B: a trendline usually suggests a (significant) trend, which there is not in this case.}\\
\\
\underline{Authors's response:} We will implement these changes.

Responses to RW2

Dear Editor,\\
\\
We are very grateful to the Reviewer 2 for the positive assessment of our paper. As detailed below in a point-by-point response to their comment, we will take into account all the remarks made during peer-review and modify the manuscript accordingly.\\
\\
Sincerely,\\
\\
Michael Hermoso\\
\\
---------\\
\\
\textit{Dear editor,}\\
\\
\textit{The work of Hermoso and Lecasble addresses the effect of seawater salinity on the oxygen isotope composition of coccoliths. This work was carried out on a variety of coccolithophores in laboratory controlled conditions (where temperature and d18O were constant). Physiological parameters and isotope composition were measured under a range of salinity. Hermoso and Lecasble have concluded that despite large physiological changes, salinity does not effect the oxygen isotope composition. This is an important observation as salinity may complicate the interpretation oxygen isotope composition in relation to sea surface temperatures.}\\
\\
\textit{Listed below are a few minor comments:}\\
\\
\textit{Line 29, Page 2 Can the authors clarify what they mean by 'synthetic salts'. Perhaps indicate the composition or recipe as this may change how an organism/cell responds to the salinity of its environment.}\\
\\
\underline{Authors's response:} We now explicitly refer to the ESAW "recipe" with reference to Keller et al. (1997), so that the chemical composition of the medium can be found.\\
\\
\textit{Line 11 and 26, Page 3 The use of the term 'bioassay' may be quite misleading. I would recommend using the term 'culture' or 'algal culture'}\\
\\
\underline{Authors's response:} We changed "biossays" for "cultures".\\
\\
\textit{Line 28, Page 3 Include a reference for the method/protocol of semi-continuous batch culture/strategy. This wouldn't be apparent to someone who is unfamiliar with the methodology and is important if someone is thinking of repeating this experiment.}\\
\\
\underline{Authors's response:} Reference to the chapter written by LaRoche et al. (2010) in the seminal book "Guide to best practices for ocean acidification research and data reporting" has been added.\\
\\
\textit{Line 7, Page 5 What do you mean by 'statistically less well behaved' (this is repeated again in Line 15, Page 7). This is quite subjective. I would suggest that the authors consider rephrasing this.}\\
\\
\underline{Authors's response:} Sentence modified for: "…with the exception of G. ericsonii, for which the measurements do not show a relation between μ and salinity."\\
\\
\textit{Line 7, Page 8 Consider including the strain names of the coccolithophores used by the different authors. This may explain the difference in the observed physiological response.}\\
\\
\underline{Authors's response:} This has been done.\\
\\
\textit{Line 14-30, Page 8 I don't think the authors can exclude the role of osmosis in determining the cell size/volume. Nor can they make the conclusion that cell size is determined by the metabolism of the organism alone. As the authors mention, there is little known about the process of osmoregulation in coccolithophores. As such it is worth considering the following:}\\
\\
\textit{1) The presence of active transporters (or membrane pumps) that may vary in type, numbers and work at different rates. This will naturally affect the transport/diffusion of water across the membrane, thus influencing the size/volume of the cell and leading to the observed difference between G. oceanica, G. ericsonii and E. huxleyi.}\\
\\

**Responses to RW2**

\textit{2) An organism may have different ways of maintaining water balance. For example, some organisms have the potential of varying their osmolytes (osmoadaptation/osmoregulation). As such, the cellular content of G. oceanica, G. ericsonii and E. huxleyi may vary based on said ability.}\\
\\
\underline{Authors's response:} We agree with the Reviewer and will simplify our text regarding these biological concepts and follow the Reviewer's first point.\\
\\
Lines 21-30 on p8 have been deleted and replaced by the following statement, which mostly relies on the Reviewer writing: "The presence of active transporters (or membrane pumps) may vary in type, numbers and work at different rates in various coccolithophore species. As a consequence, these possible distinct strain-specific features will affect the transport/diffusion of water across the membrane, thus influencing the size of the cell and leading to the observed differences between G. oceanica, G. ericsonii and the two strains of E. huxleyi."\\
\\
\textit{Line 6, Page 10 Awkward turn of phrase. Perhaps remove 'that' (which has been repeated)}\\
\\

[revised manuscript text omitted]

Changes made to Figure: % removed.